# ProstateZones – Segmentations of the prostatic zones and urethra for the PROSTATEx dataset

**William Holmlund**[1]  ⓘ                                     WILLIAM.HOLMLUND@UMU.SE
**Attila Simkó**[1]                                             ATTILA.SIMKO@UMU.SE
**Karin Söderkvist**[1]                                         KARIN.SODERKVIST@UMU.SE
**Tufve Nyholm**[1]                                             TUFVE.NYHOLM@UMU.SE
[1] *Umeå University, Department of Diagnostics and Intervention, Umeå, Sweden*

**Péter Palásti**[2]                                            PALASTI.PETER@SZTE.HU
**Szilvia Tótin**[2]                                            TOTINSZILVIA@GMAIL.COM
**Kamilla Kalmár**[2]                                           KALMAR.KAMILLA@SZTE.HU
**Zsófia Domoki**[2]                                            DOMOKI.ZSOFIA.GITTA@SZTE.HU
**Zsuzsanna Fejes**[2]                                          FEJES.ZSUZSANNA.02@SZTE.HU
**Zsigmond Tamás Kincses**[2]                                   KINCSES.ZSIGMOND.TAMAS@SZTE.HU
[2] *University of Szeged, Albert Szent-Györgyi Medical School, Department of Radiology, Szeged, Hungary*

**Patrik Brynolfsson**[1,3]                                     PATRIK.BRYNOLFSSON@SKANE.SE
[3] *Skåne University Hospital, Department of Haematology, Oncology and Radiation Physics, Lund, Sweden*

**Editors:** Accepted for publication at MIDL 2025

## Abstract

Manual segmentations are considered the gold standard for training and evaluating machine learning models in medical imaging, although difficult to obtain due to the time-consuming and labor-intensive nature of the task. We present a curated dataset of manual segmentations of the prostatic zones and intraprostatic urethra for 200 patients from the publicly available PROSTATEx dataset. For 40 patients, independent duplicate segmentations are included to provide inter-reader variability data, resulting in 240 total segmentations. The terminology follows the PI-RADS v2.1 guidelines, ensuring consistency and clinical relevance. This dataset fills a critical gap by offering a publicly available resource for training, benchmarking, and external validation of prostate MRI segmentation models.

**Keywords:** Public data, automatic segmentations, prostate, prostatic zones, urethra.

## 1. Introduction

Prostate cancer is the most common malignancy in men worldwide (Ferlay et al., 2024), requiring accurate diagnostic and treatment approaches. The Prostate Imaging Reporting and Data Systems (PI-RADS) guidelines (Turkbey et al., 2019) are used to assess prostatic lesions on multi-parametric magnetic resonance imaging (mpMRI) for diagnosis, utilizing different primary sequences depending on the zonal location. These zones are defined as the peripheral zone (PZ), central zone (CZ), transitional zone (TZ), and anterior fibromuscular stroma (AFS), which differ in characteristics and histological features (McNeal, 1988).

While radiotherapy traditionally treats the entire prostate uniformly, focal dose escalation to tumor subvolumes has shown improved outcomes (Kerkmeijer et al., 2021). However,

sparing the intraprostatic urethra is crucial to minimize urinary toxicity (Draulans et al., 2020; Leeman et al., 2022), and incorporating zonal information could enable more individualized treatment planning based on tumor location and risk profile (Ali et al., 2022).

Manual segmentation of the prostate, its zones, and the urethra is labor-intensive, thus the development of an individualized, automatic method for segmenting these structures is relevant in current medical practice. Although recent machine learning studies have had some success, they often lack standardized terminology, inter-reader comparisons, and access to public datasets for benchmarking (Wu et al., 2022).

To address these gaps, we present a publicly available dataset of 240 manual segmentations across 200 patients from the PROSTATEx dataset (Litjens et al., 2017), including all prostatic zones and the intraprostatic urethra, totaling 1200 individual structures. Forty patients include duplicate segmentations to enable inter-reader variability analysis. This dataset supports both training and unbiased evaluation of privately trained automatic segmentation models.

## 2. Method

### 2.1. Image Data

A total of 200 patients with mpMRI exams were randomly selected from the PROSTATEx dataset, available at the Cancer Imaging Archive (Clark et al., 2013). The axial T2-weighted (T2w) images were acquired with an in-plane resolution ranging from 0.3 to 0.7 mm and a slice thickness of 3.0 to 4.0 mm, with the most common resolution being $0.5 \times 0.5 \times 3.0$ mm (74%). The cohort included 66 patients with and 134 without clinically significant prostate cancer, defined as a biopsy-confirmed Gleason grade group $\geq 2$. Other image sequences are available, as well as patient selection criteria (Litjens et al., 2014).

### 2.2. Segmentations

Prostatic zones and urethra were manually segmented on axial T2w images using 3D Slicer v5.2.2 (Fedorov et al., 2012). Two experienced radiologists (>1000 and ~500 prior prostate MRI interpretations) collaborated with three junior colleagues (<100 prostate MRI interpretations each). Junior segmentations were reviewed and, if necessary, corrected by senior radiologists. The delineations were based on the zonal description by McNeal and the PI-RADS guidelines (v2.1). A retrospective dose analysis for focal boost treatment was used as the basis for the urethra, which consisted of a circular delineation with a 6 mm diameter in each slice (Groen et al., 2022). After segmenting the prostate contour and the urethra, the prostatic zones were delineated.

Forty patients were segmented independently by both senior radiologists to assess inter-reader variability, creating two segmentation sets for model testing. The remaining 160 cases had single segmentations for training. All annotations underwent quality control by a multidisciplinary team using Hero v.2023.1.1 (Hero Imaging AB, Umeå, Sweden). Minor adjustments included slice harmonization and pixel cleanup. Final approval was provided by one senior radiologist via naesView (https://naesview.com), with duplicate segmentations reviewed by the original delineator.

In the end, the resulting dataset includes 160 samples with single segmentations for training and 40 samples with duplicate segmentations for testing. All segmentations are publicly available at Zenodo (Holmlund et al., 2024).

## 3. Validation

An example of the delineated structures for a representative patient is displayed in Figure 1. For the 40 duplicate samples, inter-reader variability is presented as median with [5th, 95th]-percentiles in Table 1. The inter-reader variability metrics show overlap and boundary measures, with the Dice Similarity Coefficient (DSC) and Surface Dice. Additionally, the Center Line Distance (CLD) is provided for the urethra.

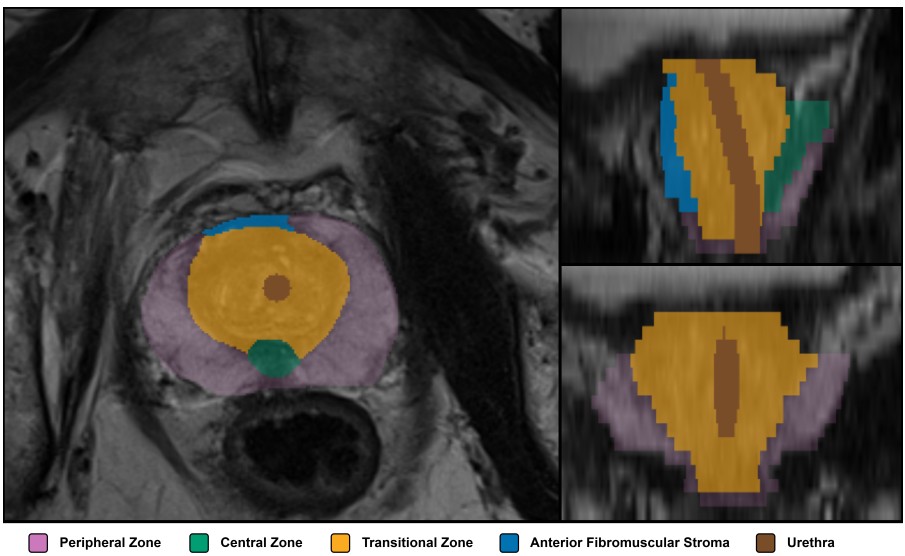

Peripheral Zone   Central Zone   Transitional Zone   Anterior Fibromuscular Stroma   Urethra

Figure 1: Example segmentation. The structures delineated for one patient displayed in the axial (left), sagittal (top right) and coronal view (bottom right).

Table 1: Inter-reader variability metrics for the 40 duplicate samples. Presented as median with [5th, 95th]-percentiles.

| Zone | DSC | Surface Dice 1 mm | Surface Dice 3 mm | CLD [mm] |
|---|---|---|---|---|
| Prostate | 0.92 [0.88, 0.94] | 0.71 [0.57, 0.78] | 0.95 [0.87, 0.98] | – |
| PZ | 0.75 [0.67, 0.83] | 0.67 [0.59, 0.73] | 0.90 [0.85, 0.95] | – |
| CZ | 0.44 [0.16, 0.63] | 0.41 [0.19, 0.58] | 0.62 [0.33, 0.84] | – |
| TZ | 0.84 [0.71, 0.90] | 0.59 [0.47, 0.66] | 0.90 [0.79, 0.96] | – |
| AFS | 0.39 [0.18, 0.54] | 0.52 [0.35, 0.66] | 0.77 [0.57, 0.89] | – |
| Urethra | 0.33 [0.16, 0.57] | 0.42 [0.23, 0.64] | 0.70 [0.47, 0.95] | 3.6 [1.9, 5.3] |

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
