# OpenReview forum: "ProstateZones – Segmentations of the prostatic zones and urethra for the PROSTATEx dataset"
_MIDL.io/2025/Short_Papers — MIDL 2025 - Short Papers_

### Official Review · Reviewer_vft3 · 2025-04-29

**Rating:** 3
**Confidence:** 5

**Summary:**

The paper presents a curated dataset of manual segmentations of the prostatic zones and intraprostatic urethra for 200 patients from the PROSTATEx dataset. It includes duplicate segmentations for 40 patients to provide inter-reader variability data, following the PI-RADS v2.1 guidelines for anatomical definitions. The dataset fills a current gap in publicly available resources for training and evaluation of prostate MRI segmentation models, offering a total of 240 segmentation sets and 1200 individual structures. The dataset is made publicly available, and the authors provide validation through reporting inter-reader metrics such as Dice scores and surface distances. The paper is clearly written, well-organized, and fits the short paper format appropriately.

**Strengths:**

- The main strength of the paper is addressing a real and relevant need in the community by providing a well-curated, standardized segmentation dataset, following clinical definitions and good annotation practices.
- The use of senior radiologists for review and quality control adds to the credibility of the annotations. Including inter-reader variability is a valuable aspect, allowing future users to benchmark model performance against human variability.
- The dataset being public and properly versioned via Zenodo is a strong contribution.
- The segmentation protocols follow established standards like PI-RADS and McNeal’s anatomical definitions, which increases clinical relevance and applicability.
- The paper’s writing is also clean, focused, and avoids unnecessary technical overcomplication, which makes it easy to follow.

**Weaknesses:**

- The main limitation is that the paper remains purely a dataset description without deeper analysis or demonstration of its impact.
- No example experiments are provided where models are trained or evaluated using this dataset, which would have strengthened the case for its usefulness.
- The inter-reader Dice scores for some structures, especially the urethra (0.33 median), are quite low, raising concerns about segmentation difficulty, but the paper does not discuss this limitation or its potential implications.
- There is also no discussion around potential challenges users may face when working with the dataset, such as resolution variability or annotation uncertainties. While useful, the contribution is incremental in nature, and there is no new algorithmic, methodological, or conceptual innovation introduced.

---

### Decision · Program_Chairs · 2025-05-01

Accept